# Adaptive Seedling Strategies in Seasonally Dry Tropical Forests: A Comparative Study of Six Tree Species

**DOI:** 10.3390/plants13202900

**Published:** 2024-10-17

**Authors:** Carlos Ivan Espinosa, Elvia Esparza, Andrea Jara-Guerrero

**Affiliations:** 1EcoSs_Lab, Departamento de Ciencias Biológicas y Agropecuarias, Universidad Técnica Particular de Loja, Loja 110107, Ecuador; akjara@utpl.edu.ec; 2Maestría de Biología de la Conservación y Ecología Tropical, Departamento de Ciencias Biológicas y Agropecuarias, Universidad Técnica Particular de Loja, Loja 110107, Ecuador; eiesparza@utpl.edu.ec

**Keywords:** biomass allocation, environmental stress, intraspecific variation, root/shoot ratio, seed size, seed germination

## Abstract

This study examines seed germination strategies and seedling establishment in six tree species typical of seasonally dry tropical forests. We focused on how interspecific and intraspecific differences in seed size and germination speed influence biomass allocation and seedling growth. Using generalized linear models, we analyzed the effects of these traits on root/shoot ratios and growth rates. Our findings reveal two main strategies: slow germination, high root/shoot ratio, and low growth rate in *Erythrina velutina* Willd and *Terminalia valverdeae* A.H. Gentry, associated with enhanced drought tolerance. In contrast, *Cynophalla mollis* (Kunth) J. Presl and *Coccoloba ruiziana* Lindau exhibited rapid germination, lower root/shoot ratios, and low to moderate growth rates, favoring competition during early establishment. *Centrolobium ochroxylum* Rose ex Rudd partially aligned with this second strategy due to its fast growth. *Vachellia macracantha* (Humb. & Bonpl. ex Willd.) Seigler & Ebinger presented a unique case, displaying slow germination and a broad range in both root/shoot ratios and growth rates. At the intraspecific level, significant variation in biomass allocation and growth rate was observed, influenced by germination speed and seed weight. We discuss the adaptive significance of seed traits in SDTFs and their role in seedling establishment under varying environmental conditions, providing insights for strategies for conservation and restoration in these ecosystems.

## 1. Introduction

Seed germination and subsequent seedling establishment are critical processes in the plant life cycle [1], playing a significant role in vegetation dynamics [2,3,4]. Among the traits influencing seedling establishment, seed size is particularly important as it links reproductive ecology to later life cycle stages, influencing competitive ability [5,6,7] and ultimately determining the distribution and abundance of plants across different habitats [8]. In harsh environments, another key trait during the early plant stages is biomass allocation [9]. A high investment in root biomass is related to drought tolerance [10,11,12]. Examining variations in seed size and biomass allocation and their relationship to seedling establishment success is key to understanding how plant communities adapt to environmental changes and predicting their potential responses to degradation and climate change scenarios [2,9,13].

The influence of seed size on plant fitness is evident through the trade-offs between fecundity and stress tolerance [14]. Seed size in plants is negatively correlated with fecundity (i.e., the number of seeds produced) [7,14] but positively correlated with germination capacity and seedling survival [15]. Species adopt different strategies that either increase fecundity or enhance stress tolerance, depending on environmental pressures [14,16,17]. Seed size, and consequently the nutritional reserves within, becomes critical in resource-limited environments, where germination and seedling establishment are particularly challenging [4]. For instance, in low-light environments with heavy litter cover, seedlings from larger seeds are better equipped to compete for resources and withstand stress [18]. Similarly, in drought-prone environments, larger seeds are more prevalent, and seedlings from these seeds demonstrate higher survival probabilities [16,18,19].

Despite theories advocating for an optimal seed size under specific ecological conditions, seed size varies widely, often spanning several orders of magnitude [8,20]. This variation is evident at multiple levels, from community-wide patterns to individual plants [7,17]. The persistence of these broad seed size ranges can be attributed to different resource allocation strategies in seedlings, which influence survival [21]. One manifestation of this optimization is the allocation of biomass between shoots and roots in response to environmental conditions [22], such as nutrient availability [19,21,23]. Under adverse conditions, biomass allocation becomes an adaptive mechanism, varying by species and ontogenetic stage [22]. For example, in Mediterranean arid ecosystems, seedling survival has been positively related to seed size and biomass allocation to roots rather than leaves and stems [19], which improves drought tolerance at the cost of reduced growth rates [24]. Intraspecific variation in seed size also significantly influences seedling establishment success. Studies on within-plant variation in seed size have shown that larger seeds tend to exhibit higher emergence rates and allocate more resources to root biomass compared to medium and smaller seeds [25,26]. These effects are often site-dependent. For example, Martínez-González et al. [26] observed higher emergence rates for large seeds in pastures but not in forests, suggesting that larger seeds are better adapted to cope with water stress and elevated temperatures. Conversely, under reduced stress conditions, the importance of the relationship between seed size, biomass allocation, and seedling survival may diminish [16]. It is well known that plants tend to allocate fewer resources to root systems when nutrient availability is high [8], investing more in foliar tissue to promote faster growth and provide a competitive advantage [24].

To evaluate the different strategies for seedling establishment, we selected model species typical of seasonally dry tropical forests (SDTF). This ecosystem is characterized by strong seasonality in precipitation, which constrains plant survival during the early life stages. Understanding these adaptive responses in seedling establishment and biomass allocation may be crucial for selecting species and seed traits in SDTF restoration efforts, where water limitation creates a bottleneck that drives high seedling mortality. 

We examined the biomass allocation and growth strategies of tree seedlings in response to environmental stress and competition at both interspecific and intraspecific levels. At the interspecific level, we investigated species differences in germination strategies, focusing on germination percentage and speed, as well as competitive strategies, defined by root/shoot ratio and seedling growth rate. At the intraspecific level, we explored how seed size and germination speed influence biomass allocation and seedling growth rate.

Our study focused on tree species typical of seasonally dry tropical forests, where plant recruitment is challenging due to a short window of approximately four months for germination and early growth before the onset of the dry season [27]. A key selective pressure for seedlings in these environments is the timing of germination; seeds that germinate later in the rainy season have less time to establish compared to those that germinate earlier [28]. However, competition is expected to be highest at the beginning of the rainy season, gradually decreasing as the season progresses [28]. Seedlings can respond to these pressures in two distinct ways. Early germination under high competition conditions may drive greater biomass allocation to shoots, resulting in a lower root/shoot ratio and faster growth rates [29]. In contrast, seedlings that germinate later in the rainy season face increased water stress and are expected to allocate more resources to root development, leading to a higher root/shoot ratio but slower growth. We hypothesized that species and individuals germinating earlier allocate more biomass to shoots to compete for light, while those germinating later allocate more biomass to roots as an adaptive response to stress tolerance. Moreover, we expect these adaptive responses to be mediated by the number of resources stored in the cotyledons or endosperm.

## 2. Results

A total of 1042 seeds from six different species were planted and monitored over a seven-month period. Germination rates varied significantly among the species. *Cynophalla mollis* exhibited the highest germination rate at 89% (from 100 seeds), followed by *Centrolobium ochroxylum* at 61% (from 242 seeds). *Coccoloba ruiziana*, *Terminalia valverdeae*, and *Erythrina velutina* showed similar germination rates of 26% (from 100 seeds), 23% (from 200 seeds), and 22% (from 200 seeds), respectively. *Vachellia macracantha* had the lowest germination rate at just 4% (from 200 seeds) (Figure 1).

### 2.1. Interspecific Germination Patterns

We observed two distinct germination patterns. The first pattern was characterized by fast and synchronous germination, as seen in *C. mollis*, *C. ochroxylum*, and *C. ruiziana*, which reached their maximum germination within the first 40 days after sowing. The second pattern exhibited extended and asynchronous germination, as observed in *T. valverdeae*, *E. velutina*, and *V. macracantha*, with germination continuing up to 120 days after sowing (Figure 1).

The generalized linear models (GLMs) revealed significant interspecific differences in seed weight, germination speed, root/shoot ratios, and growth rates among the six studied species (Figure 2). Significant differences in seed weight were observed across all species (*p* < 0.05). *C. ruiziana* had the lightest seeds (Figure 2a), while *C. ochroxylum* had the heaviest. *V. macracantha*, *C. mollis*, *E. velutina*, and *T. valverdeae* exhibited intermediate but distinct seed weights (Figure 2a). 

Germination speed varied significantly among species (*p* < 0.05). *T. valverdeae* had the longest germination time with a mean of 77 days, while *C. ruiziana* had the shortest, with a mean of 10 days. Other species, such as *C. mollis* and *C. ochroxylum*, exhibited intermediate germination speed, followed by *V. macracantha* and *E. velutina*. Although the mean germination speed for *V. macracantha* was 29 days—slightly longer than *E. velutina* at 26 days—no significant differences were detected between these two species (Figure 2b).

Significant variation in biomass allocation was also detected among species (*p* < 0.05), revealing three distinct strategies. *E. velutina* and *T. valverdeae* exhibited the highest root/shoot ratios, indicating greater investment in root biomass. Conversely, *C. mollis* allocated relatively more resources to shoots, displaying the lowest root/shoot ratios. *C. ruiziana*, *C. ochroxylum*, and *V. macracantha* exhibited intermediate levels of shoot investment. However, *V. macracantha* demonstrated a wide range of variation, similar to *T. valverdeae* and *E. velutina*, while the other species showed less variation (Figure 2c).

Growth rates also differed significantly among species (*p* < 0.05), revealing two distinct strategies: fast-growing species, such as *Centrolobium ochroxylum* and *Vachellia macracantha*, and slower-growing species. Among the slow-growing group, *Terminalia valverdeae* and *Erythrina velutina* exhibited the lowest growth rates, followed by *Coccoloba ruiziana*. Notably, *Cynophalla mollis* showed a significantly higher growth rate compared to *T. valverdeae* and *E. velutina* (Figure 2d).

Generalized linear models revealed a significant positive effect of germination speed on root/shoot ratio (*p* < 0.001), which means that delayed-germinating seeds allocated more resources to roots (Table 1). Conversely, germination speed had a significant negative effect (*p* < 0.001) on growth rate, suggesting that faster-germinating seeds invested more biomass to shoot. Seed weight had a positive and significant impact on growth rate (*p* < 0.001), with heavier seeds corresponding to greater growth rates, while the effect on the root/shoot ratio was not statistically significant (Table 1). 

### 2.2. Intraspecific Responses

At the intraspecific level, we observed more than 60% variation in seed weight across the six species. *C. ruiziana* had the lowest average seed weight (0.044 g) and the smallest range of variation (61%, 0.028–0.072 g), while *C. ochroxylum* exhibited the highest average seed weight (26.70 g) and the greatest range of variation (87%, 7.30–55.30 g) (Figure 3). Germination speed also varied widely (Table 2). *C. ruiziana* showed fast germination and the smallest variation in germination time, with only an 11-day difference between the first and last seeds to germinate. In contrast, *T. valverdeae* was the slowest species to begin germination and exhibited the second-largest variation in germination speed, following *E. velutina*.

The linear models revealed significant effects of germination speed and seed weight on the root/shoot biomass allocation ratios in some species (Table 3). In *V. macracantha*, *E. velutina*, and *T. valverdeae*, germination speed had a significant negative effect on root/shoot ratio, indicating that delayed germination resulted in reduced root allocation. Seed weights significantly affected biomass allocation only in *E. velutina* (Table 3), with a positive effect on root/shoot ratio, suggesting that heavier seeds germinating later allocated more biomass to roots. In *C. ruiziana* and *C. mollis*, neither seed weight nor germination speed significantly affected biomass allocation. In *C. ochroxylum*, where only germination speed was evaluated, no significant impact on biomass allocation was observed.

The linear models revealed contrasting effects of seed weight on seedling growth across species. Seed weight had a significant positive impact on seedling growth in *C. mollis*, while it exerted a significant negative effect on growth in *E. velutina* (Table 4). Germination speed showed a significant positive effect on growth in both *C. ochroxylum* and *E. velutina*, indicating that delayed-germinating seeds produced seedlings with higher growth rates (Table 4). In contrast, germination speed in *C. mollis* had a significant negative effect on seedling growth, with delayed germinating seeds resulting in slower-growing seedlings.

## 3. Discussion

Recruitment processes within the SDTF exhibit significant sensitivity to the strategies employed by seeds to establish themselves amidst variations in seasonal water availability and interannual fluctuations [30]. In such circumstances, intraspecific variation in seed size, as well as the strategies pertaining to biomass allocation and growth rate of seedlings, emerge as crucial determinants for the survival of numerous species [9,30]. Our findings indicate that attributes of seeds exert a notable influence on seedling establishment at both the community and population levels, albeit with varying effects across the species evaluated.

We confirmed our hypotheses that germination strategies in the SDTF are shaped by competition and environmental stress. One observed strategy involved slow germination speed, a high root/shoot ratio, and a low growth rate, which was evident in *E. velutina* and *T. valverdeae*. These findings support our hypothesis that species, particularly those facing higher stress risk—such as those with slower germination rates—allocate more resources to below-ground structures. This investment in root development likely represents an adaptive strategy to enhance drought tolerance during the dry season in seedlings characterized by low growth rates [10]. Previous studies have shown that in environments with limited water availability, seedlings that invest more in root systems tend to have higher survival rates during drought conditions [9,11,12]. This pattern is consistent with the strategy observed in *E. velutina* and *T. valverdeae*, where greater investment in root biomass likely improves access to deeper soil moisture, enhancing drought tolerance and increasing survival rates under prolonged water stress.

On the other hand, an opposite strategy was observed in *C. mollis* and *C. ruiziana*, which are characterized by rapid germination, a low root/shoot ratio, and a low to moderate growth rate. This strategy is expected in species that germinate at the beginning of the rainy season, when competition and the presence of thick leaf litter are the primary limitations for seedling establishment, particularly for smaller and less vigorous seeds [28]. *C. ochroxylum* also exhibited rapid germination and a low root/shoot ratio but differed from the other two species by displaying fast growth. Thus, the strategy of *C. ochroxylum* of low root/shoot ratio and rapid growth rate is likely supported by the nutritional reserves stored in the endosperm [31], which contribute to seedling development. 

*Vachellia macracantha* represents a unique case among the studied species. Although it shares the slow germination speed seen in *E. velutina* and *T. valverdeae*, *V. macracantha* exhibits a broad range in both root/shoot ratios and growth rates. This suggests a more plastic strategy that allows for adjustments in response to varying environmental conditions, potentially balancing between competitive ability and drought tolerance depending on resource availability [10].

Our results showed that in many of the species considered in this study, the investment in roots was relatively low compared to shoots. However, this finding aligns with the average root/shoot ratio reported for SDTFs, which is approximately 0.556 [32]. In our study, the root/shoot ratio ranged between ~0.2 and ~0.7, indicating substantial variability among species. This variation can be explained in part by differences in foliar deciduousness. We found the lowest root/shoot ratios in our evergreen species (*C. mollis*), while the highest ratios were observed in the deciduous species. This pattern is consistent with previous reports that show a mean root/shoot ratio of 0.378 for evergreen species and 0.749 for deciduous species [32]. 

At the intraspecific level, all species exhibited a wide range of variation in seed weight. This variability may provide significant ecological advantages in unpredictable environments. Such a pattern aligns with the theory of bet-hedging strategies, where producing seeds of different sizes increases the chances of survival under fluctuating environmental conditions [33]. In SDTF, larger seeds may have an advantage in coping with false starts to the rainy season or short dry spells during the rainy season [34], while smaller seeds may maximize dispersal and colonization during favorable rainy seasons. In line with our findings, previous research on *Ceiba aesculifolia* found that larger seeds have increased seedling emergence in open areas, where thermal and water stress are more pronounced, compared to within forests, while small seeds did not show significant differences in response to the site [26]. This highlights how seed size variability can influence a species’ success across different environmental gradients [17].

In our study, some species exhibited a significant effect of biomass allocation in response to germination speed. Contrary to expectations, seeds that germinated later in *E. velutina*, *T. valverdeae*, and *V. macracantha* allocated more resources to shoot biomass, resulting in a lower root/shoot ratio compared to earlier-germinating seeds. This response is counterintuitive, as later-germinating individuals would typically be expected to invest more in roots to cope with the imminent water limitations. However, two factors may explain this result. First, proportionally, these species allocate more biomass to roots compared to others (Figure 2c), allowing for greater flexibility in biomass allocation to shoots. Second, the increased shoot allocation did not result in faster growth rates (Figure 2d), suggesting that this investment may enhance the lignification of aerial structures, improving seedling survival [35]. For example, studies on transgenic plants with modified lignin profiles have shown altered biomass characteristics without affecting overall growth [36,37]. The lignification process may help seedlings withstand mechanical stresses such as wind and rain [38]. Additionally, two of these three species belong to the Fabaceae family, which is known for forming symbiotic relationships with mycorrhizae, which may increase the effective root absorption area [27,39]. The significant intraspecific variation observed in these species suggests that this adaptation is finely tuned to the timing of germination, allowing seedlings to maximize their chances of survival under varying environmental conditions. 

Interestingly, intraspecific variation in germination speed and seed weight influences seedling establishment in different ways. Seeds germinating early tended to reduce the root/shoot ratio (*E. velutina*, *T. valverdeae*, and *V. macracantha*) or modulate growth rates, either increasing (*C. ochroxylum*) or reducing them (*C. mollis* and *C. ruiziana*), thereby maximizing their chances of survival under varying environmental conditions [9,25]. Additionally, the positive effect of seed size on the growth rate of *C. mollis* and *C. ruiziana* supports established ecological theories that larger seeds, with greater nutrient and energy reserves, promote more vigorous early seedling growth [4,12]. This variability may reflect different adaptive strategies within a species [9,10], where larger seeds are more successful in stressful environments, while smaller, early-germinating seeds benefit from a longer growth period [4]. While our study includes a broad range of seed sizes from dry forest species, our findings should be interpreted with caution, as they are based on a limited number of species. Expanding the sample to include more species is necessary to fully understand the mechanisms driving adaptation in dry forest species and to inform more effective restoration efforts.

The results of this study have important implications in the context of future climate change scenarios. SDTFs are expected to experience greater variations in precipitation patterns, with more intense and prolonged drought periods [40,41], which could significantly alter seedling recruitment dynamics. Species that invest more in root biomass, such as *E. velutina* and *T. valverdeae*, may have a higher resilience to these changes, as their deeper root systems could provide access to water resources in increasingly dry conditions. In *V. macracantha*, the broad range of responses observed may allow the species to balance between competition (rapid shoot growth) and drought tolerance (increased root investment), providing resilience under fluctuating environmental conditions. This plasticity is particularly advantageous in SDTFs, where water availability is highly unpredictable. The ability to adjust biomass allocation based on resource availability might confer a survival advantage under both competitive pressure and drought stress, further underscoring the role of intraspecific variation in seed traits as an adaptive strategy to cope with environmental variability. Conversely, species that prioritize rapid shoot growth, like *C. ochroxylum*, may be more vulnerable to extended droughts, as their shallower root systems might limit access to deeper moisture during critical growth periods [34]. Therefore, intraspecific variability in seed traits may play a critical role in coping with climate changes. Understanding these adaptive strategies is essential for predicting how species will respond to future climate scenarios and for developing restoration efforts that consider increased variability in water availability [26].

In conclusion, the species studied display variations in their recruitment strategies, which are primarily mediated by germination speed. Germination speed plays a critical role in filtering processes during seed germination and seedling establishment [28]. Early-germinating species and individuals tend to allocate more resources to shoots, while later-germinating ones invest more in root development. However, this allocation strategy is mediated by seed size and the resources stored within seed structures. At the intraspecific level, the observed variation in seed size and germination speed suggests the development of diverse strategies to maximize survival in unpredictable environments.

## 4. Materials and Methods

### 4.1. Study Area and Species

Our study focused on the SDTF within the Tumbesian biogeographic region, a global hotspot of endemism that is critically threatened by extensive anthropogenic disturbances [42,43]. This region spans approximately 135,000 km^2^ along the Pacific coast of Ecuador and northern Peru, at elevations ranging from 120 to 1100 m above sea level. It is dominated by deciduous dry forests, characterized by the loss of foliage during the dry season in more than 75% of their plants [43,44]. The climate is semi-arid, with an annual precipitation ranging from 642 to 661 mm and pronounced seasonality, with about 75% of rainfall occurring between January and May [45]. The mean annual temperature ranges from 22.5 to 25.9 °C, with a markedly warmer period between December and May [45]. 

*Vachellia macracantha* (Humb. & Bonpl. ex Willd.) Seigler & Ebinger is a thorny, deciduous shrub or tree considered native to northern South America. The fruit is a 10 × 1.2 cm legume lineal, straight or somewhat curved, usually puberulent and glandular. The seed size is 5.0–6.5 × 3.6–5.3 mm [46]. *Terminalia valverdeae* A.H. Gentry is a 30 m tall tree distributed in central and southern Ecuador and extreme northwest Peru near the coast. The inflorescences are simple, from 5 to 8 cm, with white bisexual flowers. Few fruits from 2–2.3 × 6–8 cm, flattened, transversely oblong in side view, emarginate at apex and base. The seeds with two wings are fairly stiff to 2.5–3.5 × 4.3 cm [47]. *Erythrina velutina* Willd Large is a spreading tree with short-boled arms with spines. Pods ligneous from 7.5–14 × 1.2–1.7 cm irregular and deeply constricted between seeds. Usually, the pods have 1–4 red seeds with a broad black line extending from the hilum for about 3 mm. towards the chalazal end. The seed size is 14–17 mm long and 8–11 mm broad [48]. *Coccoloba ruiziana* Lindau belongs to the family Polygonaceae. It is native to Ecuador and Peru [49]. Specific information on the fruits of this species is limited. In general, species of the genus Coccoloba produce fruits eaten by a variety of birds; in the case of this species, it presents a blue/purple pseudodrupe [50]. *Centrolobium ochroxylum* Rose ex Rudd is a 6–30 m tall tree distributed at 0–350 m a.s.l. in Ecuador, west of the Andes, and the adjacent Tumbes area of Peru. Inflorescence axies, peduncles, and calyces with dense dark brown (almost black) trichomes that dry blackish brown. Flowers yellow, 15–18(-25) mm long. Fruit 10.5–27 × 4.1–10 cm, seed chamber 3–5 cm in diameter, spines 15–40 × 0.5–1 mm [31]. *Cynophalla mollis* (Kunth) J. Presl is a shrub or small tree of 2–4 m, often climbing and densely branched and leafy. Flowers are nocturnal and fragrant; sepals are orbiculate, decussate, falling before anthesis; white petals turn yellow or pale pink. Capsules 5–35 × 0.8–2 cm, linear–cylindrical, reddish-brown to yellowish-green, glabrous; seeds are 6–15 × 4–8 × 4 mm, oblongoid to ovoid-ellipsoidal, covered by a snow-white oily aril, immersed in the internal, fleshy, and juicy walls of the blood–red fruit; at maturity, the valves separate, exposing the seeds attached to the stem [51].

### 4.2. Seed Sampling and Germination

We collected fruits from three counties: Paltas, Catamayo, and Zapotillo, covering an elevation range of SDTFs from approximately 800–250 m a.s.l. This range allowed us to locate a minimum of five fruiting individuals per species. Between July 2016 and January 2017, we monitored at least five individuals per species, considering the variation in fruit production and the number of seeds per fruit. From each individual, we selected mature fruits, ensuring that only seeds free from visible pathogen or predator damage were included. For each species, we randomly selected at least 100 viable seeds to assess germination and early development. For species with low seed availability or germination rates, sample sizes were adjusted accordingly to ensure robust data. For all seeds, we measured key morphological traits, including weight (g), length (mm), width (mm), and thickness (mm). In the case of *C. ochroxylum*, the seed chamber is a thick, fibrous structure, making it impossible to remove the seeds without causing damage. Therefore, for this species, we considered the condition of the fruits as the criterion for collection, avoiding fruits with visible pathogen or predator damage. Weight and size data were recorded from the entire seed chamber rather than the seeds alone. As a result, weight was not used as a response variable in certain analyses for this species.

Seeds were disinfected in a 1% sodium hypochlorite solution for two minutes, followed by rinsing with distilled water. They were then soaked in water for 48 h. The seeds were then sown in a peat-based substrate and kept under controlled laboratory conditions with a 12 h light/12 h dark photoperiod. Watering and germination monitoring were conducted three times a week for 200 days, and germination was recorded when the radicle became visible.

Seedling growth was measured twice weekly in centimeters, from the substrate surface to the apex of the seedling. Growth monitoring continued until the seedlings developed between three and six leaves, indicating a transition to relative independence from seed reserves. At this stage, seedlings were carefully removed from the substrate, and the roots were gently washed through a sieve to minimize root loss.

The seedlings were then divided into aerial and root components by cutting at the root-shoot junction. Both parts were measured, weighed, and dried in an 80 °C oven for 48 h to determine dry biomass. The root-to-shoot dry weight ratio was calculated as an indicator of biomass allocation.

### 4.3. Data Analyses

We used the “survfit” function [52] to analyze the germination patterns of six species. We then used the Mantel–Haenszel test of the “survdiff” function, based on the G-rho family of Harrington and Fleming [53], to check for differences in the germination curves. This test includes weights on each death of S(t)p, where S(t) represents the Kaplan–Meier survival estimate. All functions used in the germination analysis are part of the survival package. 

We used a generalized linear model to assess the effect of species in four seed characteristics: weight (as a proxy of seed size), germination speed, root/shoot ratio, and growth rate (See Appendix A). For continuous variables, we applied the Gamma family error distribution due to the non-normal distribution of the data. To evaluate differences between species, we performed multiple comparisons using the “emmeans” function from the emmeans package [54] to evaluate differences between species. To evaluate the impact of seed weight and germination speed on biomass allocation and growth rate, we fitted a linear model for each species. Germination speed was measured as the number of days from planting to germination, while the growth rate was calculated using the slope of a linear regression, with growth (in centimeters) as the dependent variable and date as the explanatory variable. To assess the collinearity between seed weight and germination speed, we performed a correlation test, which confirmed no significant correlation between these variables for any species. All the analyses were carried out in the R programming environment [55].

## Figures and Tables

**Figure 1 plants-13-02900-f001:**
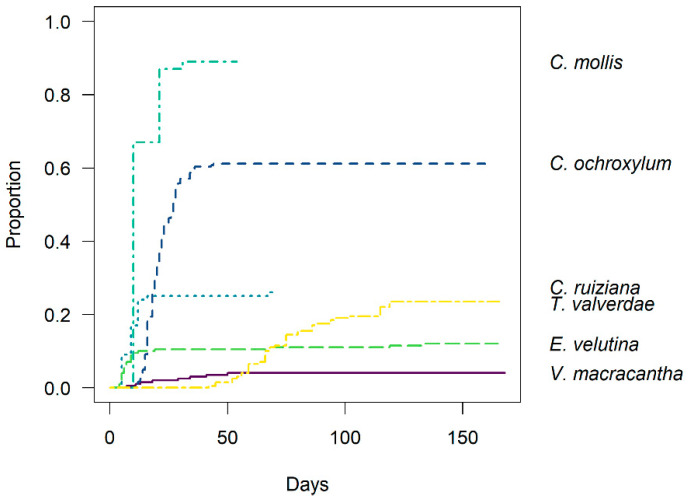
Curve of the accumulated proportion of seed germination along the time. The lines represent the proportion of seeds that germinated over time for each species. Upper curves represent higher germination proportion, and more pronounced slopes represent higher germination velocity.

**Figure 2 plants-13-02900-f002:**
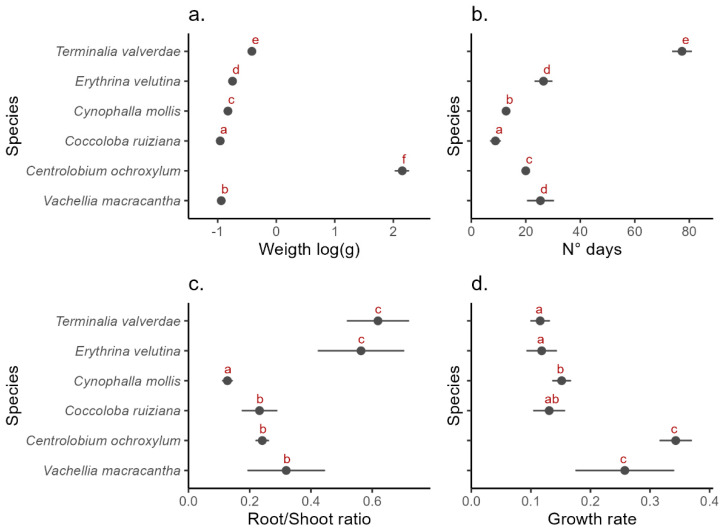
Generalized linear models of seed weight (**a**), germination speed (**b**), and competition strategies defined by biomass allocation (**c**), and growth rate (**d**). The letters show differences between species according to post hoc analysis.

**Figure 3 plants-13-02900-f003:**
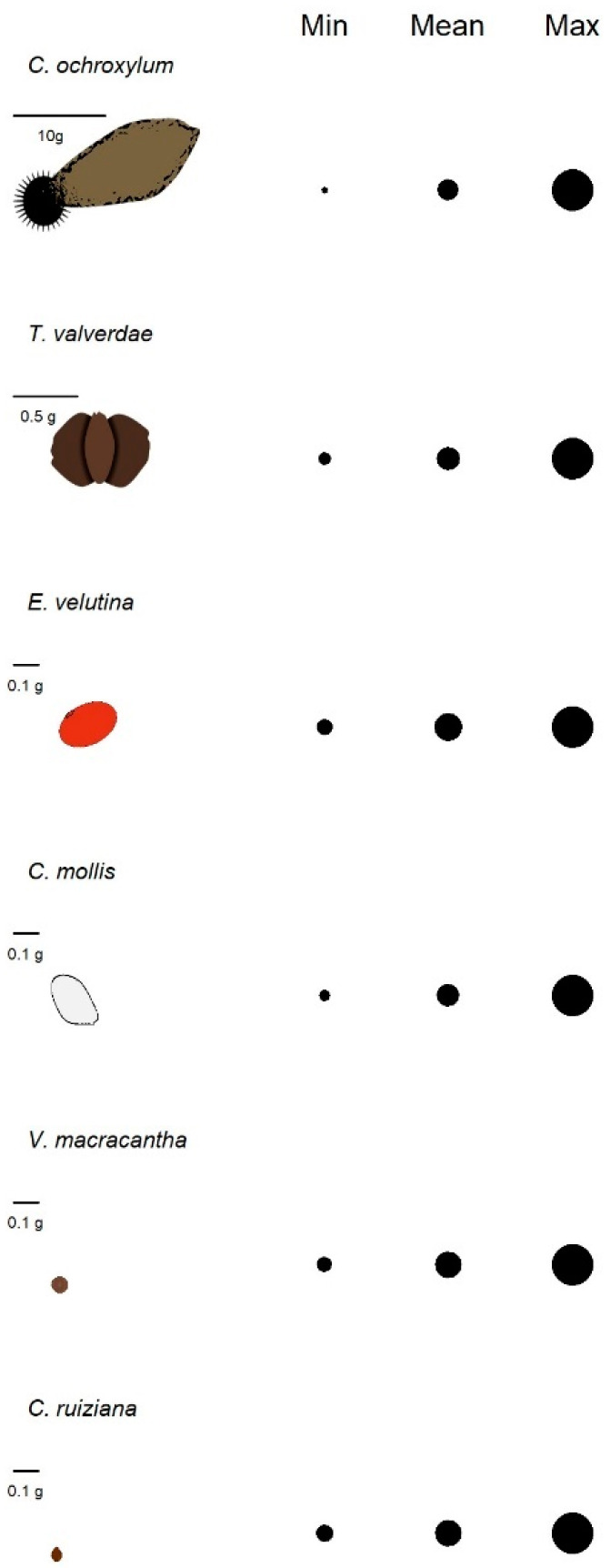
Variation in seed weight (g) of the six studied species. The size of the seed image illustrates the difference in seed weight between species, while the circles depict each species’ minimum, average, and maximum seed weight in proportion.

**Table 1 plants-13-02900-t001:** Generalized linear models assessing the effect of seed weight and germination speed (days from sowing to germination) on biomass allocation (root/shoot ratio) and seedling growth rate. The table presents the estimated coefficients, significance levels (*** *p*-value < 0.001), and standard errors (in parentheses) for each model parameter.

	Root/Shoot Ratio		Growth Rate	
Germination Speed	0.011 ***	(0.001)	−0.004 ***	(0.001)
Seed weight	0.001	(0.003)	0.027 ***	(0.002)
N	321		322	
AIC	−308.882		−687.335	

**Table 2 plants-13-02900-t002:** Variation in germination speed of the six studied species. The minimum, average, and maximum germination speed is shown in days.

Species	Min	Mean	Max
*C. ruiziana*	5	9	16
*C. mollis*	10	13	31
*C. ochroxylum*	12	20	36
*A. macracantha*	7	25	50
*T. valverdeae*	42	77	119
*E. velutina*	4	26	151

**Table 3 plants-13-02900-t003:** Linear models of the effect of seed weight and germination speed on biomass allocation (root/shoot ratio). The table presents the estimated coefficients, significance levels (*** *p*-value < 0.001; ** *p*-value < 0.01), and standard errors (in parentheses) for each model parameter.

	*V. macracantha*	*C. ochroxylum*	*C. ruiziana*	*C. mollis*	*E. velutina*	*T. valverdae*
Germination speed	−0.005 **	−0.002	−0.004	0.000	−0.004 ***	−0.006 **
	(0.001)	(0.002)	(0.006)	(0.001)	(0.001)	(0.002)
Seed weight	−0.690		0.605	−0.050	2.270 **	−0.131
	(1.347)		(4.796)	(0.099)	(0.643)	(0.206)
N	8	139	21	87	19	47
R^2^	0.855	0.007	0.033	0.003	0.721	0.177
AIC	−25.640	−239.347	−40.490	−267.666	−16.904	20.605

**Table 4 plants-13-02900-t004:** Linear models of the effect of seed weight and germination speed on seedling growth rate. The table presents the estimated coefficients, significance levels (*** *p*-value < 0.001; ** *p*-value < 0.01; * *p*-value < 0.05), and standard errors (in parentheses) for each model parameter.

	*V. macracantha*	*C. ochroxylum*	*C. ruiziana*	*C. mollis*	*E. velutina*	*T. valverdae*
Germination speed	0.001	0.006 **	−0.002	−0.002	−0.000 *	0.000
	(0.001)	(0.002)	(0.001)	(0.001)	(0.000)	(0.000)
Seed weight	1.260		−0.276	0.339 ***	−0.493 *	0.024
	(1.779)		(1.023)	(0.089)	(0.175)	(0.031)
N	8	140	21	87	19	47
R^2^	0.182	0.069	0.129	0.185	0.428	0.041
AIC	−21.190	−198.096	−105.383	−286.968	−66.277	−158.195

## Data Availability

The data that support the findings of this study are available from the corresponding author upon reasonable request.

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
