# Peer review of "Adaptive Seedling Strategies in Seasonally Dry Tropical Forests: A Comparative Study of Six Tree Species"

_plants, 2024, doi:10.3390/plants13202900_

Round 1

Reviewer 1 Report

Comments and Suggestions for Authors

The manuscript titled Adaptive Seedling Strategies in Seasonally Dry Tropical Forests: A Comparative Study of Six Tree Species.” provides valuable insights into the adaptive seedling strategies of six tree species in seasonally dry tropical forests (SDTFs). The research examines how variations in seed size, germination speed, and biomass allocation affect the growth and survival of seedlings in environments characterized by limited water availability and competition. The findings contribute significantly to understanding the role of seed traits in seedling establishment, with implications for conservation and restoration strategies in these threatened ecosystems. I believe that the manuscript needs major revisions given in the attached file.

Comments on the Quality of English Language

The manuscript is generally well-written and clear, but there are some areas where sentences can be restructured for better readability. For instance, long sentences should be broken down to avoid complexity and ensure a smoother flow of ideas.

There are occasional inconsistencies in verb tenses, particularly when transitioning between discussing past research and presenting current findings. Ensure that the past tense is consistently used when referring to prior research, and the present tense is used for current findings.

Author Response

We appreciate all comments made on the manuscript. We have reviewed and incorporated each suggestion. The responses to each comment are included in detail in the attached document.

Reviewer 2 Report

Comments and Suggestions for Authors

This is a study in which authors examined seed germination strategies and seedling establishment in some species from seasonally dry tropical forests aiming to make a relationship among seed size, germination speed, biomass allocation and seedling growth. It is a very good article and I have no doubt about its publication. The discussion concerning the adaptive significance of seed traits and their role in seedling establishment in response to environmental conditions, as well as the way seed traits mediate plant responses to competition are very important. However, it is so important that authors make a final comment concerning the limits of their findings since the number of species and populatinos is a little bit limited. Also, they should avoid to cut the table between two pages.

Author Response

Comments 1: This is a study in which authors examined seed germination strategies and seedling establishment in some species from seasonally dry tropical forests aiming to make a relationship among seed size, germination speed, biomass allocation and seedling growth. It is a very good article and I have no doubt about its publication. The discussion concerning the adaptive significance of seed traits and their role in seedling establishment in response to environmental conditions, as well as the way seed traits mediate plant responses to competition are very important. However, it is so important that authors make a final comment concerning the limits of their findings since the number of species and populatinos is a little bit limited. 

Response 1: We appreciate the revisor comments. To clarify the limits of our findings, in the discussion, we have pointed out that, "although our study includes a broad range of seed sizes from dry forest species, our findings should be interpreted with caution, as they are based on a limited number of species. Expanding the sample to include more species is necessary to fully understand the mechanisms driving dry forest species adaptation and inform more effective restoration efforts" (Lines: 289-293).

Comment 2: Also, they should avoid to cut the table between two pages.
Response 2: We have organized the paragraphs and spacing to prevent the tables from being split between two pages.

Round 2

Reviewer 1 Report

Comments and Suggestions for Authors

The manuscript titled “Adaptive Seedling Strategies in Seasonally Dry Tropical Forests: A Comparative Study of Six Tree Species” presents valuable insights into the mechanisms of seedling establishment and survival in challenging environments. By focusing on germination speed, biomass allocation, and growth strategies among six tree species, the study significantly contributes to our understanding of plant adaptation in Seasonally Dry Tropical Forests (SDTF). These findings have important implications for biodiversity conservation and ecological restoration efforts, especially as climate change exacerbates environmental stress in such ecosystems. I believe that the manuscript needs major revisions.

Minor Comments to the Authors:

  1. Clarify Methodological Details: The description of the seedling measurement process in the "Materials and Methods" section could be enhanced by providing additional details regarding the precise intervals of monitoring. Including this would improve the clarity of the seedling growth measurement procedures.
  2. Expand on Intraspecific Variation: The section discussing intraspecific variability in seed traits could benefit from more in-depth discussion of how these variations influence long-term survival under fluctuating environmental conditions. Providing further examples from the literature would strengthen the argument.
  3. Table and Figure Enhancements: Ensure consistency in font size and labeling across all tables and figures. For example, Figure 2's axis labels are smaller compared to the others. Standardizing the presentation would improve overall readability.
  4. Statistical Significance Reporting: In Table 1, it would be helpful to explain the practical implications of the statistical significance (p-values) observed in your generalized linear models. This would provide readers with a clearer understanding of the ecological relevance of your results.
  5. Germination Patterns Discussion: The discussion of the germination patterns could explore further the potential environmental cues that might influence these observed strategies. This could include factors such as soil moisture levels or light availability that interact with the germination speed of different species.

Author Response

Comment 1: The description of the seedling measurement process in the "Materials and Methods" section could be enhanced by providing additional details regarding the precise intervals of monitoring. Including this would improve the clarity of the seedling growth measurement procedures.

Answer 1: We have addressed this by specifying that watering and germination monitoring were conducted three times a week for 200 days, with germination recorded as soon as the radicle became visible (L: 388-390).

The details regarding seedling measurements are presented in lines 391-395.

Comment 2: The section discussing intraspecific variability in seed traits could benefit from more in-depth discussion of how these variations influence long-term survival under fluctuating environmental conditions. Providing further examples from the literature would strengthen the argument.

Answer 2: We have thoroughly revised and expanded the discussion on intraspecific variation, providing a deeper analysis to address the reviewer's suggestion, as well as improving the use of references. In the former version of our manuscript, we already empathized on the importance of variability in seed weight at the intraspecific level, particularly in relation to the bet-hedging strategy. This strategy enables species to cope with unpredictable conditions, where larger seeds offer advantages during short dry spells, while smaller seeds enhance dispersal and colonization in more favorable conditions (L: 255-260). Additionally, in this new version, we have incorporated an example from a study on Ceiba aesculifolia, which demonstrates how larger seeds outperform smaller ones in open areas with higher stress, but not within forest sites, thereby highlighting the ecological advantages of seed size variability across different habitats (L: 262-266).

In the former version, we also expanded the discussion on germination speed and its influence on biomass allocation, particularly focusing on the unexpected finding that later-germinating seeds in Erythrina velutina, Terminalia valverdeae, and Vachellia macracantha allocated more biomass to shoots. This may enhance lignification of aerial structures, supporting seedling survival despite reduced root allocation (L: 272-280). In this new version, we have extended the discussion on the implications of intraspecific variation in biomass allocation in V. macracantha, particularly in the context of future climate change scenarios (L: 306-313).

Comment 3: Ensure consistency in font size and labeling across all tables and figures. For example, Figure 2's axis labels are smaller compared to the others. Standardizing the presentation would improve overall readability.

Answer 3: We have reviewed and standardized the font size and labeling across all figures and tables. Specifically, we have modified the labels in Figure 1 to ensure consistency and improve overall readability.

Comment 4: In Table 1, it would be helpful to explain the practical implications of the statistical significance (p-values) observed in your generalized linear models. This would provide readers with a clearer understanding of the ecological relevance of your results.

Answer 4: We have revised the paragraph discussing the results of the generalized linear models to better explain the ecological implications of our findings (L: 152-158). Additionally, the legend for Table 1 has been updated to specify that germination speed refers to the number of days from sowing to germination, and we have included explanations of p-values represented by asterisks to facilitate the interpretation of the results.

Comment 5: The discussion of the germination patterns could explore further the potential environmental cues that might influence these observed strategies. This could include factors such as soil moisture levels or light availability that interact with the germination speed of different species.

Answer 5: As the reviewer noted, the discussion primarily focuses on germination patterns to contextualize seedling development strategies, rather than making germination a central point. While we recognize that environmental cues such as soil moisture and light availability are relevant, our primary focus is on seedling strategies rather than germination patterns alone. As a result, we have chosen not to expand this section significantly, to avoid diverting attention from our core argument.